# The Functions of the Demethylase JMJD3 in Cancer

**DOI:** 10.3390/ijms22020968

**Published:** 2021-01-19

**Authors:** Anna Sanchez, Fatma Zohra Houfaf Khoufaf, Mouhamed Idrissou, Frédérique Penault-Llorca, Yves-Jean Bignon, Laurent Guy, Dominique Bernard-Gallon

**Affiliations:** 1Department of Oncogenetics, Centre Jean Perrin, CBRV, 63001 Clermont-Ferrand, France; anna.sanchez@etu.uca.fr (A.S.); houfaf.fz@gmail.com (F.Z.H.K.); mouhamed.idrissou@etu.uca.fr (M.I.); Yves-Jean.BIGNON@clermont.unicancer.fr (Y.-J.B.); 2INSERM U 1240 Molecular Imagery and Theranostic Strategies (IMoST), 63005 Clermont-Ferrand, France; Frederique.PENAULT-LLORCA@clermont.unicancer.fr (F.P.-L.); lguy@chu-clermontferrand.fr (L.G.); 3Department of Biopathology, Centre Jean Perrin, 63011 Clermont-Ferrand, France; 4Department of Urology, Gabriel Montpied Hospital, 63000 Clermont-Ferrand, France

**Keywords:** JMJD3, cancers, H3K27me3, epi-drugs

## Abstract

Cancer is a major cause of death worldwide. Epigenetic changes in response to external (diet, sports activities, etc.) and internal events are increasingly implicated in tumor initiation and progression. In this review, we focused on post-translational changes in histones and, more particularly, the tri methylation of lysine from histone 3 (H3K27me3) mark, a repressive epigenetic mark often under- or overexpressed in a wide range of cancers. Two actors regulate H3K27 methylation: Jumonji Domain-Containing Protein 3 demethylase (JMJD3) and Enhancer of zeste homolog 2 (EZH2) methyltransferase. A number of studies have highlighted the deregulation of these actors, which is why this scientific review will focus on the role of JMJD3 and, consequently, H3K27me3 in cancer development. Data on JMJD3’s involvement in cancer are classified by cancer type: nervous system, prostate, blood, colorectal, breast, lung, liver, ovarian, and gastric cancers.

## 1. Introduction

At present, cancer is a major cause of death worldwide. There were 18.1 million new cases and 9.6 million deaths caused by cancer in 2018. About one in five men and one in six women will develop the disease during their lifetime [1].

The reasons behind the development of cancer are many and varied. Cancer can be sporadically caused by a spontaneous mutation in the DNA sequence, and the risk factors cannot be determined with certainty. There are a number of other risk factors that can lead to the formation of cancer. We classify these factors into two groups: intrinsic factors, including age, hormonal status, and family history and extrinsic factors affecting the individual’s environment and lifestyle, such as diet, physical activity frequency, and hormonal contraceptive use (National Cancer Institute, Bethesda in Maryland (MD), United States). Epigenetics are influenced by external or internal environmental factors, such as the specific characteristics of the cell, and play a major role in oncogenesis by modulating chromatin compaction and, consequently, the expression of some genes involved in tumor development [2].

Epigenetics can be defined as the modification of gene expression without modification of the DNA sequence. These epigenetic variations are reversible and transmissible to one’s descendants. The epigenetic mechanisms involved are the action of non-coding RNAs (ncRNAs), such as small RNAs, micro RNAs (miRNAs), PIWI-interacting RNAs (piRNAs), endogenous short RNAs (endo-siRNAs), and long non-coding RNAs (lnRNAs)) [3]. DNA methylation and the covalent post-translational modifications (PTMs) of histones, among which acetylation and methylation are the most studied. Epigenetic modifications contribute to changes in chromatin compaction and are widely studied in cancerology [4,5,6].

Epigenetic mechanisms are controlled by the action of actors capable of reading, erasing, or writing epigenetic modifications [7,8]. This review focuses on the involvement of the jumonji domain-containing protein 3 demethylase (JMJD3)—whose role is to demethylate the di- and tri-methylation of lysine 27 from histone 3 (H3K27me3), a repressive epigenetic mark—in various cancers [9,10].

## 2. Results

### 2.1. Post-Translational Modification of Histones

PTMs of histones are known to lead to changes in the structure of chromatin [11]. Chromatin is composed of a repetition of nucleosomes, which contain ~147 pb of DNA wrapped around an octamer of histones (2H2A, 2H2B, 2H3, and 2H4), wherein H1 has the role of locking the nucleosomal structure. Chromatin is composed of DNA, small histone basic proteins, and non-histone proteins involved in replication and transcription. The nucleosome is the primary level of DNA condensation and the fundamental unit of chromatin’s structure [7,12].

Histones are basic proteins that are highly conserved during evolution and have three α helices—a structure called a “histone fold”. These α helices are connected to each other by loops called L1 and L2 [13]. These basic proteins have a *C*-terminal tail, located in the center of the histone octamer, and an *N*-terminal tail that is often enriched with lysine (K) and arginine (R), which can undergo PTMs. PTMs of the *N*-terminal histone tails allow for the regulation of gene expression by modifying chromatin compaction and access to DNA by transcription factors [11].

PTMs are varied and include acetylation, methylation, phosphorylation, ubiquitination, sumoylation, ADP ribosylation, and many others. These PTMs are controlled by different actors, such as histone acetyltransferases (HATs) and histone deacetylases (HDACs), which, respectively, place or remove an acetyl from the tails of histones. Histone methyltransferases (HMTs) and histone demethylases (HDMs) add or remove one or more methyl groups [11].

The hypoacetylation and hypermethylation of histone tails are often associated with chromatin closure and low transcriptional activity, while histone tail hyperacetylation and hypomethylation are usually associated with chromatin opening and the high transcription of genes [12]. The acetyl group neutralizes the positive charge of lysine and causes a lower interaction with the negative charge of DNA phosphates. This results in relaxation of the chromatin [13].

The constant “cross-talk” between the different epigenetic marks positioned in the genome at a given time leads to real complexity in the understanding of epigenetic modifications. This, in turn, has a different impact on the gene expression, integrity, and stability of the genome [7,14].

### 2.2. JMJD3 Demethylase

Demethylase JMJD3, also known as lysine-specific demethylase 6B (KDM6B) belongs to the JmjC-containing protein family, members of which are highly conserved during evolution [15], with activity dependent on α-ketoglutarate and oxygen and requiring Fe(II) as a cofactor [9,16]. The JMJD3 gene is located on chromosome 17p13.1 [17] and has 88% homology with the gene encoding the ubiquitously repeat transcribed tetratricopeptide repeat on the X chromosome (UTX) [18]. UTX is also capable of removing the di-trimethylation of H3K27 [9,16,18,19].

H3K27me3 is known to be a repressive epigenetic mark deposited thanks to the SET catalytic domain of Zeste Homolog 1 methyltransferase activator (EZH1) and Enhancer of Zeste Homolog 2 (EZH2) when combined with the Polycomb repressive complex 2 (PCR2). It is notably found in facultative heterochromatin [10] and is read by the Polycomb repressive complex 1 (PCR1) which ubiquitinates a histone H2A lysine that allows the recruitment of factors leading to chromatin compaction and consequently a repression of gene expression [20]. The demethylating activity of JMJD3 at H3K27me2/3 makes it a transcription activating protein [21].

JMJD3 demethylase has been widely reported in embryonic development processes. JMJD3 allows the activation of WNT3 via the demethylation of H3K27me3 at the beginning of endoderm differentiation and activation of Dickkopf-related protein 1 (DKK1) at the end of endoderm differentiation [22]. The inactivation of JMJD3 in mice causes respiratory failure, leading to perinatal lethality. JMJD3’s activity is essential for maintenance of the pre-Bötzinger complex because the demethylase controls some key regulatory genes of the complex (Reln, March4, Kirrel3, and Esrrg) [23].

Several studies linked the involvement of JMJD3 in osteogenesis, which indirectly activates the HOX and BMP genes, thereby promoting osteogenic differentiation [24]. In addition, JMJD3 plays a regulatory role in HOX expression during axial structuring in mice [25]. The transcription factor promyelocytic leukaemia zinc finger (PLZF) is essential for the osteogenic differentiation of human mesenchymal stem cells (hMSCs). During the pre-osteoblast differentiation stage, the recruitment of JMJD3 during gene coding for PLZF, coupled to H3K27ac’s enrichment of the locus, contributes to PLZF overexpression [26]. JMJD3 is also a key regulator of the cartilage maturation process during enchondral ossification by promoting the expression of Runt-related transcription factor 2 (RUNX2), an essential transcription factor for chondrocyte maturation and by enhancing osteoblast differentiation and, therefore, skeletal ossification [27,28].

JMJD3 appears to have a significant role in hematopoiesis, in association with vitamin C, and demethylase appears to be essential in hematopoietic stem cell production in zebrafish and humans. [29]. Also in a zebrafish model, JMJD3 was shown to be important in early myelopoiesis regulation by promoting the expression of Spi1, an essential transcription factor for myelopoiesis in vertebrates [30].

A few studies shed light on the correlation between JMJD3 and immunity: H3K27me3 demethylation by JMJD3 or UTX allows the expression of specific genes responsible for the terminal differentiation of immune T cells. For example, one of these genes, Sphingosine-1-Phosphate Receptor 1 (*S1PR1*) codes for a sphingosine receptor, which is essential to the release of differentiated cells from the thymus and Kruppel Like Factor 2 (*KLF22*) coding for a transcription factor involved in the late maturation of T cells [31].

In 2007, de Santa et al. demonstrated a link between inflammation and epigenome reprogramming. A strong induction of JMJD3 was observed in mouse macrophages after stimulation by lipopolysaccharides (LPSs), acting during inflammation [32]. In 2012, the same observation was made after the catalytic activity of JMJD3 was inhibited by GSK-J1. JMJD3 induction was directly regulated by the NF-kB transcription factor, an essential factor in gene inflammation induction [33]. More recently, JMJD3 activity inhibition by its chemical inhibitor ethyl 3-((6-(4,5-dihydro-1*H*-benzo[*d*]azepin-3(2*H*)-yl)-2-(pyridin-2-yl)pyrimidin-4-yl)amino)propanoate (GSK-J4) showed a decline in Natural Killer (NK) activities, immune cells that have a pro-inflammatory effect. H3K27 demethylation induced by JMJD3 is important for NK cell function [34].

Neuronally, JMJD3 is involved in the survival of differentiated neurons from the hippocampus and is also important in regulating the maturation of granular cerebellar neurons in mice. This makes the transcription of genes such as Tiam1 and Grin2c able to regulate the Brain-Derived Neurotrophic Factor (BDNF), an important regulator of brain development [35].

JMJD3 demethylase also has a role independent of demethylation. Data show that JMJD3 acts in chromatin remodeling independently of its demethylase activity through interaction with transcription factors such as T-bet, T-box, and the SWI/SNF remodeling complex [36,37]. In addition, JMJD3 has been shown to have a role in cellular reprogramming. JMJD3 inhibits the cell reprogramming mediated by transcription factor Oct4 by targeting PHF20 independently of its demethylase activity. PHF20 is then ubiquitylated via an E3 ligase and is directed to the proteasome to be degraded [38].

### 2.3. JMJD3 and Cancer

A number of studies have highlighted the link between JMJD3 and tumor progression. However, its role in oncology is still unclear—is there a balance between oncogenic or tumor suppressor activity in JMJD3?

#### 2.3.1. Nervous System Cancer

##### Glioblastoma

The brain is the organ for which JMJD3 has been the most studied. Among different cancers that affect the brain, glioma or glioblastoma are those for which we have collected the most data.

Glioblastoma is the most common and aggressive carcinoma of the central nervous system. In children with pediatric brainstem glioma, a common mutation in the *H3F3A* gene that codes for a variant of the histone H3.3 has been observed. This mutation leads to PCR2 sequestration and a decrease in H3K27me3, as well as more significant disease progression. Methylation maintenance by inhibiting JMJD3 has been shown to be a therapeutic strategy for glioma treatment [39,40]. JMJD3 is overexpressed in glioma tissue relative to healthy tissue. The inhibition of demethylase activity, and hence the maintenance of transcriptional gene repression, resulted in decreased cell proliferation and migration, as well as increased apoptosis [41]. These results were later confirmed after *JMJD3* inhibition by Clustered Regularly Interspaced Short Palindromic Repeats Interference (CRISPRi), which still resulted in a decrease in glioma cell proliferation, migration, and invasion, making *JMJD3* an oncogene involved in the epithelial–mesenchymal transition (EMT). JMJD3 is also able to increase EMT marker expression, such as the expression of snail homolog 1 (*SNAI1)* [42], whose overexpression leads to an increase of *N*-cadherins and fibronectin proteins by regulating the C-X-C motif chemokine 12 (CXCL12) at the transcriptional and translational levels [43].

By contrast, some studies highlighted *JMJD3* as a tumor suppressor gene in glioblastoma. Most of these genes, including the p53 transcription factor, are known to have a role in senescence and DNA repair, inhibit angiogenesis, and thus, act as a tumor suppressor actor [44]. *JMJD3’*s methylation state has been demonstrated to be essential for its function [45]. A study showed that JMJD3 is able to regulate p53 independently of chromatin modification but directly via interaction with p53. This leads to glioblastoma stem cell (GSC) differentiation, which suppresses GSC proliferation [46]. Interestingly, JMJD3 is a direct target of Signal transducer and activator of transcription 3 (STAT3). STAT3 binds to the *JMJD3* promotor, resulting in the promotor’s repression. The overexpression of *JMJD3*, after STAT3 inhibition, indicates a decrease in neurosphere formation, cell proliferation, and, therefore, tumorigenicity. JMJD3 regulation by STAT3 was found in tumor samples from different patients [47]. These results together with those of Ene et al. show the importance of JMJD3’s role in GSCs differentiation and GSC proliferation [46,47]. Receptor tyrosine kinase (RTK) overexpression, such as that due to platelet-derived growth factor receptor A (PDGFRα), plays an important role in the regulation of cell growth and survival. PDGFRα is amplified to 5–10% in glioblastoma [48]. By targeting PDGFRα with an aptamer of its ligand, the PDR3 has been shown to decrease STAT3 expression and increase JMJD3 expression. Aptamer treatment also resulted in an increase in P53 in response to increased JMJD3 expression [49]. This result places JMJD3 in a central position between STAT3 and p53 in limiting tumor progression.

##### Medulloblastoma

Among brain cancers, medulloblastoma (MB) is a cancer that mostly develops in children (9.2% of brain tumors) between 0 and 14 years of age, with a peak of incidence during the first decade [50]. JMJD3’s role is controversial in this type of pathology. It was first demonstrated in 2014 in vitro and in vivo on mice that JMJD3 acts in favor of carcinogenesis by modulating bivalent chromatin domain expression. These domains contain the repressive mark H3K27me3 and the active mark H3K4me3. Gene expression in these regions remains inactive but awaits activation. In association with Shh signaling, JMJD3 demethylates H3, activating Shh target genes, such as GliA. These genes play a predominant role in the proliferation of cerebellum granule neuron precursors (CGNPs) in the early postnatal period. Moreover, the Shh MB subtype features low H3K27me3 accumulation, suggesting that JMJD3 is more active in tumors compared to normal cerebellar tissues and that JMJD3 demethylase activity inhibition by shRNA in Shh MB subtypes leads to a decrease in CGNP proliferation and thus a decrease in growth MB [51].

Conversely, analyses were performed on tumor and healthy human samples in 2017. It was demonstrated via immunohistochemistry that the expression of topoisomerase IIβ (topoIIβ), a protein essential for DNA repair, was inversely correlated with the repressive mark H3K27me3 in MB tumors with low topoIIB expression. JMJD3 expression was higher in normal tissues than in MB tissues, making JMJD3 a protein involved in topoIIB expression maintenance and helping to limit tumor progression [52].

##### Neuroblastoma

Neuroblastoma are cancers that develop from the neural crests responsible for the sympathetic nervous system and adrenal gland formation and account for 8–10% of all tumors in children. *MYCN* gene amplification is highly relevant to the loss of differentiation potential and the poor prognosis associated with this disease [53]. JMJD3 inhibition in neuroblastoma has been shown to be favorable to carcinogenesis reduction by modulating the expression of key genes involved in the formation of neuroblastoma and notably by reducing MYCN proto-oncogene expression and overexpressing the p53 and the p53 upregulated modulator of apoptosis (PUMA) factors involved in apoptosis. GSK-J4 inhibitor treatment has been shown to be effective in combination with retinoic acids, leading to a more effective reduction in cell viability, as well as the induction of cell differentiation and reticulo-endoplasmic stress, which leads to cell death [54].

#### 2.3.2. Prostate Cancer

Presently, 1,276,106 men worldwide have been diagnosed with prostate cancer (PC), and 358,989 deaths are estimated. After lung and colorectal cancer, prostate cancer is the third most common cancer among men [1].

Increasingly more studies are highlighting the implications of epigenetic alterations in prostate tumor progression. Several studies highlighted EZH2 overexpression in PC [55,56]. EZH2 methyltransferase inhibition results in decreased proliferation, suggesting that decreased H3 methylation has a protective effect on the cancer development. Therefore, JMJD3’s involvement as an H3 demethylase appears to be essential to limit tumor progression [55]. On the other hand, it seems surprising that EZH2 and JMJD3 are concomitantly overexpressed during PC development [55,57]. A study by Xiang and colleagues discussed this phenomenon: PC is a heterogeneous cancer [9] with different degrees of aggressiveness (Gleason score (GS)). The two proteins EZH2 and JMJD3 may act differently depending on the type of PC. Moreover, several studies have revealed this phenomenon, according to which there is a differential overexpression of EZH2 and JMJD3 depending on the aggressiveness of the disease [56,58,59]. A global genome-wide analysis of H3K27me3 was performed on prostate biopsies representing three clinical groups (healthy, GS ≤ 7, and GS > 7), and a strong representation of H3K27me3 was found in tumor tissue compared to healthy tissue, with a positive correlation according to the GS [60]. Another study showed the impact of H3K27me3 on the regulation of the Retinoic acid receptor β2 (*RARβ2*), estrogen receptor (*Erα*), progesterone receptor (*PGR*), and Repulsive Guidance Molecule A (*RGMA*) genes and also correlated H3K27me3 with the level of tumor aggressivity [57]. These genes are involved in PC development. The coverage of these previous genes by JMJD3 and EZH2 is greater in PC cell lines than in healthy cells with larger JMJD3 coverage at the these gene promotors [56].

PC is a hormone-dependent cancer and is particularly androgen-dependent. Thus, understanding the link between androgen receptor (AR) and PC development is essential to better treat this disease. Like many genes, AR expression is modulated by epigenetic mechanisms. AR is overexpressed in PC, as well as in 30% of castration-resistant PC (CRPC) compared to untreated PC [61]. A number of studies have suggested a possible link between JMJD3, H3K27me3, and the AR metabolic pathway [56,57,61,62,63,64]. JMJD3’s transcriptional level is increased in LNCaP AR+ tumor cell lines compared to PC-3 AR− tumor cell lines [56]. This comment also applies to JMJD3’s enrichment on four candidate genes (O—methylguanine-DNA methyltransferase (*MGMT)*, transformer 2 alpha homolog (*TRA2A)*, 2 small nuclear RNA auxiliary factor 1 (*U2AF1)*, and ribosomal protein S6 kinase A2 (*RPS6KA2)*), identified as signature genes in PC [59]. After GSK-J4 inhibitory treatment of JMJD3, AR− tumor cells did not show an increase in JMJD3 on the promotor of candidate genes, while AR+ tumor cells did show an increase [62]. This observation was echoed at the transcriptional level on the same genes after inhibitor treatment [56]. The relevance of androgens in PC and epigenetics was reported by Sun et al.: The repressive cluster miR99a/let7c/125-b that contributes to a decrease in cell growth is regulated by androgens and by the couple, EZH2/JMJD3. JMJD3 expression induces an increase in miRNA cluster expression giving it a tumor suppressor role [65].

While Varambally et al.’s work showed that EZH2 inhibition resulted in reduced cell proliferation [55], a study conducted on AR-WT and AR-∆LBD tumor cells (CRPC cell lines) showed that JMJD3 inhibition by GSK-J4 also resulted in reducing PC cell proliferation [61]. This confirms the complexity of the epigenetic mechanisms and the need to balance understanding of JMJD3/EZH2 in PC.

One study proposed a model in which JMJD3 would have an oncogenic effect. JMJD3’s inhibition by Triptolide, extracted from a medicinal herb, would have an anti-tumor effect on PC tumor cells. This compound would induce an increase in H3K27me3 after JMJD3 inhibition and an increase in H3K9me3 by positively regulating the SUV39H1-HP1α complex, which is involved in heterochromatin formation. The increase in these two repressive marks would result in gene transcription suppression, as well as an increase in senescence [66].

#### 2.3.3. Blood Cancer

There are a wide variety of blood disorders, including T-cell acute lymphoblastic leukemia (T-ALL), acute myeloid leukemia (AML), Hodgkin’s lymphoma (HL), and diffuse large B-cell lymphoma (DLBCL), also known as Non-Hodgkin’s lymphoma. These various disorders are caused by the deregulation of a number of signaling pathways in which JMJD3, in some cases, also appears to be involved.

##### T-ALL

This pathology affects both children and adults and is due to a deregulation of thymocyte T growth and maturation, wherein Notch receptor 1 (NOTCH1) is overexpressed in 60% of cases. NOTCH1 is implicated in the oncogenic pathways. In 2014, JMJD3 was identified as a central player in oncogene activation mediated by NOTCH1. JMJD3 was found to be overexpressed and colocalized with NOTCH1 in T-ALL cells. Ubiquitin-specific peptidase 7 (USP7) stabilizes the complex NOTCH1/JMJD3 through deubiquitination. The NOTCH1 oncogenic targets are controlled by this mechanism in controlling the demethylation of target genes through JMJD3 activity. It was demonstrated that concomitant USP7 and JMJD3 inhibition lead to a reduction of tumor growth in vivo. [67]. JMJD3 was found at the promotor site of NOTCH1 targets that have oncogenic functions: *HES1*, *HEY1*, *NRARP* [68]*, c-myc,* and Notch receptor 3 (*NOTCH3*) [69]. This allows T-ALL cell initiation and maintenance by controlling the methylation of oncogenic targets [68]. These studies suggest that JMJD3 is an interesting therapeutic target to reduce T-ALL progression. Other researchers have examined JMJD3’s functions in hematopoietic stem cells (HSCs) that are not related to chromatin modification. As in T-ALL cells, JMJD3 is overexpressed in HSCs, and JMJD3 loss upregulates the transcription factor AP-1 and increases the downstream expression of *Jun* and *Fos,* resulting in the inability of HSCs to self-renew after stress. In a T-ALL model transduced with a retrovirus expressing the NOTCH1 intra-cellular domain, the authors showed that JMJD3 loss did not produce greater survival, unlike the AML model. The initiation and maintenance of T-ALL and AML by JMJD3 would depend on the cell subtype and developmental context. JMJD3 could be an interesting target if its inhibition does not affect normal HSCs but only those implicated in leukemia [70].

##### AML

On the other hand, JMJD3, when induced by all-trans retinoid acid (ATRA) signaling, is also known to have an oncorepressive role as it acts on targets such as C/EBPβ and RIPK3 (a key suppressor of AML malignancy) that allow the myeloid differentiation, cell cycle arrest, and cell death of AML cells [71]. In the context of differentiation therapy, JMJD3 has also been shown to allow malignant cells to differentiate into treatable differentiated cells. In AML, Aurora kinase (AURK) A represses JMJD3 expression, thus favoring leukemogenesis and justifying the predominant role of JMJD3 in the repression of AML progression [72].

In contrast, other studies demonstrated the oncogenic role of JMJD3 in AML development [73,74,75]. JMJD3 overexpression in AML cells is associated with a poor prognosis, and JMJD3 inhibition facilitates an increase in H3K27 methylation associated with a decrease in proliferation and *HOX* expression in vitro, as well as a decrease in disease progression in vivo. [75]. Illiano et al. demonstrated a link between the cAMP response element-binding protein (CREB), which plays a critical role in leukemogenesis, and JMJD3 by using GSK-J4 treatment on AML cell lines. The use of GSK-J4 permits the downregulation of CREB. The latter protein is phosphorylated on Ser-133 by a protein kinase A (PKA) before being directed to the proteasome [73]. Moreover, it seems that the association of GSK-J4 with a natural cAMP raising compound (forskolin) potentiates the anti-proliferative effects of GSK-J4 via PKA in some AML cells (U937 cells), leading to B-cell lymphoma 2 protein (BCL2) downregulation and caspase 3 activation [74].

##### Lymphoma

Whether in HL [76] or DLBCL [77,78], JMJD3 is over-expressed in germinal center B (GC-B) cells. JMJD3 is modulated by viruses such as the Epstein–Barr virus (EBV), which is involved in the development of GC-B and the development of HL. EBV and the latent oncoprotein membrane protein-1 (LMP1), modified in 50% of HLs by EBV, induce strong regulation of JMJD3 in primary B cells. The NOTCH homolog 2 *N*-terminal-like (NOTCH2NL) and CD58 genes have been identified as JMJD3 targets in GC-B and are overexpressed in HL cells. After JMJD3 inhibition by interfering RNA, a reduction of H3K27me3 is observed on target genes, which provides sufficient evidence that JMJD3 is involved in the development of HL via LMP1 in GC-B [76].

In DLBCL, JMJD3 overexpression in GC-B is associated with a poor prognosis. Demethylation, moreover, plays a role in disease development. JMJD3 affects the B-cell receptor (BCR) and the downstream B-cell lymphoma 6 protein (BCL6). These proteins are involved in B-cell survival, and the use of the JMJD3 inhibitor significantly sensitizes B cells to chemotherapy treatments compared to the use of treatments alone [77]. In both subtypes of DLBCL, which are the ABC or GCB subtypes, overexpressed JMJD3 acts on the regulatory factor 4 interferon (IRF4), which plays a role in hematopoietic cell differentiation. On the other hand, two distinct mechanisms have been put forward depending on DLBCL type: The ABC subtype seems to use a pathway linking JMJD3–IRF4–NF–kB to induce cell survival, while for the GCB subtype, survival is mediated by the Bcl-2 survival gene, which is sensitive to JMJD3 demethylase activity [78]. There also appears to exist a connection between JMJD3 demethylase and cyclin-dependent kinase 9 (CDK9), which is abnormally expressed in DLBCL cells. CDK9 inhibitors utilization affects JMJD3 and H3K27 methylation leading to a weakening of the expression of anti-apoptotic myeloid cell leukemia 1 (MCL1) and X-linked inhibitor of apoptosis (XIAP) proteins [79].

However, in anaplastic large cell lymphomas (ALCLs), an oncogenic mutation, such as in the *N*-terminal portion of nucleophosmin and in the intracellular domain of anaplastic lymphoma kinase (NPM-ALK), in cells may lead to a weak expression of the p16INK4a/Rb tumor suppressive pathway. This pathway activates oncogene-induced senescence (OIS), which limits tumor development. JMJD3 is found to be a partial contributor to the activation of p16INK4a/Rb in an NPM-ALK context and could be a novel agent to contribute to the upregulation of the p16INK4a/Rb pathway [80].

#### 2.3.4. Colorectal Cancer

Colorectal cancer (CRC) is the second most deadly cancer in the world [1], but relatively few investigations address the contribution of JMJD3 to the development of CRC.

Once again, JMJD3 presents itself as a double actor. It is under expressed in CRC on clinical samples [81] and acts as a tumor suppressor when activated by the active metabolite of vitamin D: 1,25(OH)_2_D_3_. Although the mechanism remains unclear, JMJD3’s presence limits the expression of key participants implied in EMT (*SNAI1*, Zinc finger E-box-binding homeobox 1 (*ZEB1)*) and facilitates chromatin access to the genes involved in cell adhesion and the 1,25(OH)_2_D_3_ target genes (*CYP24A, CDHA1/E-cadherin,* and *CST5/cystatin D*), all of which have a role in restraining tumor progression [82,83]. In Tokugunaga et al., the loss of the JMJD3 was considered a marker of poor prognosis in CRC and promoted cell proliferation, suppression of apoptosis, and reduction in the expression of the p15INK4b tumor suppressor [81]. JMJD3 also regulates tumor immunity in CRC by enabling the transcription of the Th-1 type chemokine (*CXCL9* and *CXCL10*), and treating the primary cells of CRC with GSK-J4 leads to a reduction in the chemokine level. These chemokines have been shown to be positively correlated with CRC patients’ survival [84].

On the other hand, JMJD3 plays a role in the histone promotor demethylation of the epithelial cell adhesion molecule (EpCAM) gene playing a role in the proliferative process in some of cancers and which is overexpressed in CRCs [85]. In addition, the aberrant activation of NOTCH1 in CRC, specifically serrated neoplasia pathway, leads to an increase in Ephrin type-B receptor 4 (EPHB4), which promotes tumor growth and cell migration. It turns out that the intracellular NOTCH domain accompanies JMJD3 from the cytosolic compartment to the nuclear one to modify EPHB4 chromatin architecture in CRC [86]. These latest data position JMJD3 as a CRC oncoprotein.

#### 2.3.5. Breast Cancer

Breast cancer (BC) is the leading cause of death in women worldwide [1]. There are different subtypes of this cancer depending on the expression of a number of receptors [87]. On the other hand, hormone-dependent BCs may become resistant to hormonotherapy, hence the interest in finding new therapeutic strategies. Among those strategies, epigenetics is increasingly revealing itself as an interesting area of research in the discovery of new targets for anti-cancer treatments.

Svotelis et al. showed that after stimulation of Erα by estrogens, JMJD3 is co-recruited at the Erα binding regions on the promotor of the B-cell lymphoma 2 (*BCL2*) anti-apoptotic gene to induce its expression. The link between the modification of BCL2’s chromatin architecture and its constitutive activation in estrogen-resistant cancer cells has been demonstrated, identifying JMJD3 and methyltransferase EZH2 as interesting therapeutic targets. Indeed, estrogen-resistant cancer cells overexpress HER2, which results in the constitutive activation of *BCL2* by inactivating EZH2 through the AKT pathway. In this case, the involvement of JMJD3 is no longer needed, which highlights the existence of non-genomic pathways in the regulation of the *BCL2* gene by EZH2, whereas JMJD3 is able to positively regulate *BCL2* expression in a genomic way in non-resistant breast cancer cell lines [88]. JMJD3 could be targeted in nonresistant cancer cells, and EZH2 in both nonresistant cancer cells and estrogen-resistant cancer cells, to limit *BCL2* activation.

It is clear that JMJD3 has a role on cancer stem cell (CSC) pluripotency. In BC, JMJD3 has an inhibitory effect on Oct4 (stemness marker) via the degradation of PHF20 in triple-negative and luminal BC cell models [89]. In both luminal and triple-negative models, JMJD3 acts in disfavor of stemness characteristics. The same finding was reported by Zhao et al. on fibroblasts [38]. However, a further study highlighted the favorable effect of inhibiting JMJD3 in limiting tumor progression in vitro and in vivo with a decrease in cell proliferation and pluripotency markers (Nanog, Sox2, and Oct4) and in tumor size and weight after GSK-J4 treatment [90]. In addition, as previously seen in CRC [82,83], Xun et al. confirmed that JMJD3 is activated by vitamin D analogues such as paricalcitol (vitamin D receptor agonist), also resulting in decreased stem-cell-like properties [89]. Controversially, JMJD3 is involved in promoting the mesenchymal characteristics induced by TGF-β and activating *SNAI1* in BC [91]. An additional study showed that the microenvironment plays a role in the activity of JMJD3 and, particularly, hypoxia within tumors through a reduction in the activity of demethylase and, consequently, a greater presence of the H3K27me3 marker at the epigenome [92]. While Wei et al. demonstrated that a loss of H3K27 trimethylation is associated with a poor predictor in BC [93], another team highlighted the link between hypoxia, JMJD3, and the development of stem cell capacity by regulating the methylation state of DICER. The lack of oxygen leads to a reduction of JMJD3 oxygen-dependent demethylase activity that reduces DICER expression because of its over-methylated state. DICER controls miRNA production and DICER repression, leading to a repression of the miR-200 family that activates EMT [92,94]. In this context, JMJD3 activity seems to be important to counter the development of stem cell capacity and EMT, thus reducing metastasis.

#### 2.3.6. Lung Cancer

JMJD3’s function in lung cancer is controversial. In the cell model of adenocarcinomic human alveolar basal epithelial cells (A549 cell line), JMJD3 contributes to the proliferation of A549 by affecting the cell cycle (decreasing p21 mRNA production), but has no effect on cell migration [95]. Conversely, a recent study showed its effects on EMT activation and its favorable influence on cell migration in a non-small cell lung cancer (NSCLC) model [96]. The same study also demonstrated an elevation in the expression of JMJD3 in the tumor tissue and showed a positive correlation with cancer progression to lymph node metastasis. These results are consistent with those found in primary sarcomatoid carcinoma of the lung, which is a type of pulmonary sarcomatoid carcinoma where JMJD3 is inversely correlated with patient survival [97]. This is of interest as a study performed on human serum samples from blood (healthy or NSCLC-affected) showed a decrease in the JMJD3 mRNA expression of serum NSCLC positively correlated with disease severity (N1 and N2 metastasis nodes) [98]. This differential expression of JMJD3 between serum and tumor tissue could potentially establish JMJD3 as a diagnostic tool and biomarker for NSCLC.

#### 2.3.7. Other Types of Cancer

The table below summarizes and describes how JMJD3 is regulated in previously described cancers and liver, ovarian, and gastric cancer and what the consequences are in terms of its role in favor of, or against, tumor progression (Table 1).

### 2.4. Epi-Drugs of JMJD3

As previously highlighted in this review, epi-drugs are increasingly evolving towards new therapeutic strategies against cancer to regulate cancer progression. In the table below (Table 2.), we listed a few molecules likely to target JMJD3 in cancer and other pathologies.

GSK-J4 is the most studied molecule related to JMJD3 targeting and is an ester derivative of the GSK-J1 molecule. GSK-J4 is able to penetrate cells more easily and is hydrolyzed to GSK-J1, which binds to the active site of JMJD3 competitively with α-ketoglutarate [33].

A team recently developed pan-histone demethylase inhibitors based on the chemical characteristics of tranylcypromine (a known inhibitor of Lysine-specific demethylase 1 (LSD1) belonging to the superfamily of the flavin adenine dinucleotide (FAD)-dependent amine oxidases), with the scaffolds 2,2′-bipyridine or 5-carboxy-8HQ, two competitive groupings of α-ketoglutarate. However, these compounds cannot be described as specific for JMJD3 since they simultaneously inhibit LSD1 and enzymes with a jumonji catalytic domain [110]. In addition, seven compounds derived from quinoline-5,8-dicarboxylic acid have been shown to exert strong inhibition on JMJD3 by binding to its active JMJD3 site [111]. Very recently, Giordono et al. also demonstrated the inhibitory effect of a molecule derived from 2-Benzoxazol-2-yl-phenol. One of these compounds (compound 8), moreover, showed cell cycle arrest on melanoma cancer cells [112]. Further, flavonol analogs (myricetin) were modeled, and eight of them appeared to have a strong affinity for the active site of JMJD3 compared to the binding of GSK-J1. On the other hand, these molecules have not been tested in vitro or in vivo [113]. Finally, Zhang et al. identified two new inhibitors (salvianic acid A and puerarin 6″-O-xyloside) using a capillary electrophoresis technique [114]. However, further studies at the cellular level should be done to validate the inhibitory effect of these molecules on JMJD3 (Table 2).

To conclude, there is no doubt that JMJD3 is involved in carcinogenesis. We note that demethylase frequently acts on deregulated pathways in cancers (EMT, cell cycle, and P53). On the other hand, it is difficult to categorize JMJD3 as a tumor suppressor or oncoprotein. We highlighted in this scientific review the heterogeneous tissue-dependent demethylase activity of JMJD3. Within the same organ, the aggressiveness of the disease seems to influence the role of JMJD3. It is well known that epigenetic modifications are complex and that there is permanent context-dependent crosstalk between different epigenetic marks. It would be interesting to better understand this epigenetic dialogue in cancers to identify more precisely the role of JMJD3 in cancers. In addition, molecules (Table 2) capable of modifying JMJD3 activity have been developed. Unfortunately, apart from GSK-J4 and GSK-J1, these molecules are still rarely or never used experimentally, but they could become future epidrugs for clinical trials, treatments, and therapies for cancer patients.

## Figures and Tables

**Table 1 ijms-22-00968-t001:** Jumonji domain-containing protein 3 demethylase regulation in liver, ovarian, and gastric cancers ^a^.

Types of Cancer	Up/Down Regulation	How?	Oncogene/ Tumor Suppressor	References
Liver	Upregulated	+ EMT + Migration + Invasion via slug demethylation	Oncogene	[99]
	Downregulated by miR-941 in hepato carcinoma cell lines	− EMT − Invasion − Migration − Proliferation	Tumor Suppressor	[100]
Ovarian	-	+ Cell death + Self-renewal + Tumor initiation	Oncogene	[101]
	Upregulated	+ EMT + Migration + Invasion + Proliferation + Metastatic Capacities via TGF-β1 expression	Oncogene	[102]
	Upregulated in drug resistant ovarian cancer cells, positively correlated with HER2 expression	+ HER2 expression	Oncogene	[103]
Gastric	Upregulated	− Survival + Proliferation	Oncogene	[104]
	-	+ p15/p16 when associated with Long noncoding RNAs (ARHGAP27P1)	Tumor suppressor	[105]
Glioblastoma	Upregulated	+ Tumor growth + Proliferation − Apoptosis + Migration + Invasion + EMT (*SNAI1* and fibronectin and *N*-cadherins by regulating CXCL12)	Oncogene	[39,40,41,42,43]
		+ P53 − Neurosphere formation − Proliferation	Tumor suppressor	[46,47,49]
Medulloblastoma	Upregulated	+ Proliferation (+ shh signaling)	Oncogene	[51]
	Downregulated	+ DNA repair (+ topoIIβ)	Tumor suppressor	[52]
Neuroblastoma	-	+ Differentiation (− *MYCN*) + Apoptosis (p53, PUMA) + Cell death (+ reticulo-endoplasmic stress)	Tumor suppressor	[53,54]
Prostate cancer	Upregulated (Differential expression according to AR status and aggressiveness)	+ Proliferation + Senescence	Oncogene	[61,66]
		− Proliferation + DNA repair (+ *MGMT*)	Tumor suppressor	[55,57,59,62,65]
T-ALL	Upregulated	+ NOTCH1 target genes + Tumor growth + AP-1	Oncogene	[67,68,69,70]
AML	-	+ Differentiation + Cell cycle arrest + Cell death (+ C/EBPβ and RIPK3)	Tumor suppressor	[71,72]
	Upregulated	+ Proliferation + Disease progression + *HOX* cAMP/PKA	Oncogene	[73,74,75]
HV	Upregulated	JMJD3 activated by LMP1 + NOTCH2NL	Oncogene	[76]
DLBCL	Upregulated	+ Survival (BCR, BCL6, IRF4-NF-kB, BCL2)− Apoptosis (XIAP, MCL1)	Oncogene	[77,78,79]
ALCL	-	+ p16INK4a/Rb pathway	Tumor suppressor	[80]
	-	+ Proliferation (+ EpCam) + Tumor growth +Migration (NOTCH1/JMJD3 → +)	Oncogene	[85,86]
Colorectal cancer	Downregulated	JMJD3 activated by Vitamin D active metabolite − EMT (− *SNAI1*, − *ZEB1*, + cell adhesion genes) − Proliferation + Apoptosis + P15INK4B + Survival (+*CXCL9*, *CXCL10*)	Tumor suppressor	[81,82,83,84]
Breast cancer	-	− Survival (+ *BCL2*) + Proliferation + Pluripotency (Nanog, Sox, Oct4) + Tumor growth + EMT (JMJD3 activated by TGFβ → + *SNAI1*)	Oncogene	[88,90,91,93]
	-	− Pluripotency (− Oct4)− Stem-like properties − EMT (+DICER → − miR-200)	Tumor suppressor	[89,92,94]
Lung cancer	Upregulated	+ Cell cycle (− p21) + EMT (Migration) + Metastasis − Survival	Oncogene	[95,96,97,98]

^a^ EMT: Epithelial-mesenchymal transition; CXCL12: C-X-C motif chemokine 12; topoIIβ: Topoisomerase iiβ; SNAI1: Snail homolog 1; ZEB1: Zinc finger E-box-binding homeobox 1; Topoisomerase iiβ; MGMT: O—methylguanine-DNA methyltransferase; NOTCH1: Notch receptor 1; CREB: cAMP response element-binding protein; PKA: Protein kinase A; LMP1: Latent membrane protein-1; NOTCH2NL: Notch homolog 2 N-terminal-like; BCR: B-cell receptor; BCL6: B-cell lymphoma 6 protein; IRF4: Regulatory factor 4 interferon; BCL2: B-cell lymphoma 2 protein; XIAP: X-linked inhibitor of apoptosis; PUMA: p53 upregulated modulator of apoptosis; MCL1: Myeloid cell leukemia 1; EpCam: Epithelial cell adhesion molecule; JMJD3: Jumonji Domain-Containing Protein 3 demethylase.

**Table 2 ijms-22-00968-t002:** Mechanisms of jumonji Domain-Containing Protein 3 demethylase (JMJD3) epi-drugs.

Epi-Drugs	Mechanisms	References
GSK-J4	Hydrolyzed to GSK-J1 Competitive fixation	[33,34,39,54,67,68,69,74,75,90,101,104,106,107,108,109]
Pan-histone demethylase inhibitor	Covalent binding of tranylcypromine fragments and reversible binding of jumonji domain demethylase inhibitor scaffolds. Competitive fixation	[110]
Quinoline-5,8-dicarboxylic Acid Derivative	Fragmentary approach to quinoline-5,8 dicarboxylic acid Competitive fixation	[111]
2-Benzoxazol-2-yl-phenol Scaffold (Compound 8)	Fragmentary approach to 2-Benzoxazol-2-yl-phenol Competitive fixation	[112]
Flavonol analogues (myricetin)	Competitive fixation	[113]
Salvianic acid A and puerarin 6″-O-xyloside	Identified by capillary electrophoresis	[114]

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
