# Peer review of "The Functions of the Demethylase JMJD3 in Cancer"

_ijms, 2021, doi:10.3390/ijms22020968_

Round 1

Reviewer 1 Report

The review by Sanchez et al. “The functions of the demethylase JMJD3 in cancer” is interesting, deals with current topic and would be of potential interest for IJMS readers. However, a number of issues have to be addressed prior accepting for publication in the journal.

  1. Lanes 33-35: Epigenetics is also influenced by intrinsic factors, which must be reflected in the statement.
  2. Lane 38: Epigenetic mechanisms are not restricted to non-coding microRNAs, what about long non-coding RNAs etc.?
  3. Each abbreviation should be explained in text, when used for the first time e.g. PTMs, MLL, HL etc.
  4. Lane 44, 176: the role of JMJD3 is not suppressing but demethylation
  5. Lane 59: actors or factors?
  6. Lane 65-67: the sentence is not correct
  7. Lane 78: the reference does not support the statement exactly
  8. Lane 79: too many “repeat”
  9. Lane 82: what do you mean by “trademark registered”
  10. Lane 111-113: The sentence should be somewhat introduced
  11. Lane 116: proteins?
  12. Lane 135: Correct the sentence
  13. Lane 171: Reference 48 shows slightly different age and %
  14. Lane 178-181: the sentence is confusing
  15. Lane 201: Reference 8 refers reference 53
  16. Lane 208: Later on what?
  17. Lane 225: Reference 62 = 55!
  18. Lane 244-245: rather repressive marks?
  19. Lane 254-257: the sentence is confusing
  20. Lane 263-265: It is not clear to me: JMJD3 deregulates AP-1 (dimer of fos and jun) and in parallel increases the expression of Fos and Jun?
  21. Lane 277: prognosis?
  22. Lane 278-280: the sentence is not completed
  23. Lane 282: There should be reference 72
  24. Lane 291-293: Check the sentence please
  25. Lane 322-324: The sentence is confusing
  26. Lane 351-354: Add reference
  27. Lane 367-370: what do you mean by the expression of JMJD3 in serum?
  28. Table 1: Ovarian: use “Cell death”
  29. Lane 379-380: the sentence is not completed
  30. Lane 381: I do not think that GSK-J4 is the most studied molecule of all.
  31. Lane 395-396: which team?

Author Response

Please, to follow the right lane, refer to the word version with active track changes.

Moreover, our manuscript is ready to be correct to your MDPI English editing service.

Reviewer 1

  1. Lanes 33-35: Epigenetics is also influenced by intrinsic factors, which must be reflected in the statement.
  2. Lane 38: Epigenetic mechanisms are not restricted to non-coding microRNAs, what about long non-coding RNAs etc.?
  3. Each abbreviation should be explained in text, when used for the first time e.g. PTMs, MLL, HL etc.
  4. Lane 44, 176: the role of JMJD3 is not suppressing but demethylation
  5. Lane 59: actors or factors?
  6. Lane 65-67: the sentence is not correct
  7. Lane 78: the reference does not support the statement exactly
  8. Lane 79: too many “repeat”
  9. Lane 82: what do you mean by “trademark registered”
  10. Lane 111-113: The sentence should be somewhat introduced
  11. Lane 116: proteins?
  12. Lane 135: Correct the sentence
  13. Lane 171: Reference 48 shows slightly different age and %
  14. Lane 178-181: the sentence is confusing
  15. Lane 201: Reference 8 refers reference 53
  16. Lane 208: Later on what?
  17. Lane 225: Reference 62 = 55!
  18. Lane 244-245: rather repressive marks?
  19. Lane 254-257: the sentence is confusing
  20. Lane 263-265: It is not clear to me: JMJD3 deregulates AP-1 (dimer of fos and jun) and in parallel increases the expression of Fos and Jun?
  21. Lane 277: prognosis?
  22. Lane 278-280: the sentence is not completed
  23. Lane 282: There should be reference 72
  24. Lane 291-293: Check the sentence please
  25. Lane 322-324: The sentence is confusing
  26. Lane 351-354: Add reference
  27. Lane 367-370: what do you mean by the expression of JMJD3 in serum?
  28. Table 1: Ovarian: use “Cell death”
  29. Lane 379-380: the sentence is not completed
  30. Lane 381: I do not think that GSK-J4 is the most studied molecule of all.
  31. Lane 395-396: which team?

Lane 33-36: Epigenetics are influenced by external or internal environmental factors, for example, the specific characteristics of the cell, and plays a major role in oncogenesis by modulating chromatin compaction and consequently the expression of some genes involved in tumor development [2].

Lane 39-43: other small RNAs action were added.

action of non-coding RNAs (ncRNAs) such as small RNAs (micro RNAs (miRNAs), PIWI-interacting RNAs (piRNAs) and endogenous short RNAs (endo-siRNAS) or long non-coding RNAs (lnRNAs)) [3], DNA methylation and covalent post-translational modification (PTMs) of histones, among which acetylation and methylation are the most studied. Epigenetic modifications contribute to changes in chromatin compaction and are widely studied in cancerology [4–6].

HL abbreviation is at the right position lane 326. PTMs abbreviation has been modified lane 42.

Lane 48: suppress replaced by demethylate

Lane 63: actors replaced by factors

Lane 69-71: The sentence was clarified.

 Hypoacetylation and hypermethylation of histone tails are often associated with chromatin closure and low transcriptional activity, while histone tail hyperacetylation and hypomethylation are usually associated with chromatin opening and high transcription of genes

Lane 74-76: re writed sentence to clarify

Lane 84: a reference was changed

Lane 83-88: re writed sentence to clarify and not repeat

Lane 89: trademark registered modified by mark deposited

Lane 100: DDK1 abbreviation was added

Lane 107: PLZF abbreviation was added

Lane 111: RUNX2 abbreviation was added

Lane 121 : the sentence was introduced

Lane 125: LPS are not protein but lipopolysaccharides acting during inflammation. This was modify.

Lane 144-146: re writed sentence as JMJD3, a balance between oncogenic or tumor suppressor activity?

Lane 148: Glioblastoma for title

Lane 161: added Clustered Regularly Interspaced Short Palindromic Repeats Interference before (CRISPRi)

Lane 164: SNAI1 abbreviation was added

Lane 173: STAT3 abbreviation was added

Lane 183-187: sentences were clarify in removing the referring to different authors (Sherry Lynes and Ene et al.) and a conclusion was added.

Lane 190: medulloblastoma for title

Lane 191-193: changed 3 to 10 in 0 to 14 according the reference

Lane 198: changed suppresses to demethylate

Lane 201-205: Sentence was clarify.

. Moreover, the Shh MB subtype has a low H3K27me3 accumulation suggesting that JMJD3 is more active in tumors compared to normal cerebellar tissues and JMJD3 demethylase activity  inhibition by shRNA in Shh MB subtypes leads to a decrease in CGNPs proliferation and thus a decrease in growth MB [51].

Lane 214: add Neuroblastoma for title

Lane 230: suppressed reference 9

Lane 238: later on replaced by moreover

Lane 255: references was corrected. There was a problem in Zotero.

Lane 256 -259: MGMT U2AF1 TRA2A and RPS6KA2 abbreviations were added

Lane 278 : markers replaced by marks

Lane 282: Abbreviation HL was here

Lane 288-289: It was precise that NOTCH1 is implicated in oncogenic pathways

Lane 288-295: the sentence has been clarify: Ubiquitin-specific peptidase 7 (USP7) stabilizes the complex NOTCH1/JMJD3 through deubiquitination. The NOTCH1 oncogenic targets are controlled by this mechanism in controlling demethylation of target genes through JMJD3 activity. It was demonstrated that concomitant USP7 and JMJD3 inhibition conducted to a reduction of tumor growth in vivo.

Lane 301-308: information was added to better understand with a conclusion:

As in T-ALL cells, JMJD3 is overexpressed in HSCs and JMJD3 loss leads to upregulate the transcription factor AP-1 and increases the downstream expression of Jun and Fos resulting in an inability of HSCs to self-renew after stress. In a T-ALL model transduced with a retrovirus expressing NOTCH1 intra-cellular domain, authors showed that JMJD3 loss did not produce greater survival unlike AML model. The initiation and maintenance of T-ALL and AML by JMJD3 would depend on cell subtype and developmental context. JMJD3 could be an interesting target if its inhibition does not affect normal HSCs but only those implicated in leukemia [70].

Lane 314-316: Conclusion added: thus favoring leukemogenesis and justify of the predominant JMJD3 role in the repression of AML progression

321-323: The sentence is completed with : 

 Illiano et al, demonstrate a link between cAMP response element-binding protein (CREB), which plays a critical role in leukemogenesis, and JMJD3 by using GSK-J4 treatment on AML cell lines

Lane 325-327: Information about anti proliferative effect on AML cells were added: Moreover, it seems that the association of GSK-J4 with a natural cAMP raising compound (forskolin) potentiates the anti-proliferative effects of GSK-4 via PKA in some AML cells (U937 cells) leading to a BCL2 downregulation and caspase 3 activation

Lane 319: Diagnosis replaced by prognosis

Lane 321-323: the sentence was completed: Illiano et al, demonstrate a link between cAMP response element-binding protein (CREB), which plays a critical role in leukomogenesisleukemogenesis, and JMJD3 by using GSK-J4 treatment on AML cell lines.

Lane 325 : the reference was changed

Lane 326 : lpm1 corrected in lmp1

Lane 333: NOTCH homolog 2 N-terminal-like were added before NOTCH2NL

Lane 353: anaplastic large cell lymphomas (ALCLs) were added

Lane 356-358: conclusion added: JMJD3 is found to be a partial contributor to the activation of p16INK4a/Rb in an NPM-ALK context and could be a novel agent to contribute to upregulate the p16INK4a/Rb pathway

Lane 369-373: The sentence was clarify: Not surprisingly, as in glioma [43], JMJD3 also regulates tumor immunity in CRC by turning the transcription of Th-1 type chemokines (CXCL9, CXCL10) permissive, whereas in CRCand treated primary cells of CRC with GSK-J4 leads to a reduce in chemokines level. These chemokines have been shown to be positively correlated with CRC patients’ survival , chemokines are diminished [84].

Lane 377-380: Ephrin type-B receptor 4 added before  (EPHB4 )and EBPH4 corrected by EPHB4 (lane 380 also)

Lane 384 to 388: sentence has been clarify: Indeed, estrogen-resistant cancer cells overexpress HER2 that result in a constitutive activation of BCL2 by inactivating EZH2 through AKT pathway. In this case, the involvement of JMJD3 is no longer needed and this highlight the existence of non-genomic pathway in the regulation of BCL2 gene whereas JMJD3 is able to positively regulate BCL2 expression in a genomic way in non-resistant breast cancer cell lines  

Lane 392-396: Information have been added to clarify and conclude

Lane 398: It is precise that Oct4 is a stemness marker

Lane 410: reference 90 was add

Lane 398-400: A conclusion has been add about stemness characteristics in breast cancer.

In both luminal and triple-negative models, JMJD3 act in disfavor of stemness characteristics.

412-419: Sentences were added to clarify and conclude

 While Wei et al. demonstrated that a loss of H3K27 trimethylation is associated with a poor predictor in BC [93], another team highlighted the link between hypoxia, JMJD3 and the development of stem cell capacity by regulating the methylation state of DICER. The lack of oxygen leads to a reduction of JMJD3 oxygen-dependent demethylase activity that reduces DICER expression because of its over-methylated state. DICER is controlling miRNA production and DICER repression consequently conducts to a repression of miR-200 family that activates EMT [92,94]. In this context, JMJD3 activity seems to be important to counter the development of stem cell capacity and EMT thus reduces metastasis.

Lane 427: it is precise that the serum samples come from blood samples and lane 429 it was precised that authors looked JMJD3 mRNA expression in serum samples.

Lane 428 to 430: In response to the question “what do you mean by the expression of JMJD3 serum: I precise JMJD3 mRNA lane 429. Authors worked with blood samples and centrifuged samples to obtain serum. Analysis have been done on JMJD3 mRNA expression in serum samples.

Lane 437: Table 1 has been modified with all type of cancer event those described in the text.

Table 1: Cell Death replaced death cell

Lane 342-343: the sentence was completed: We list here in the table below (Table 2.) a few molecules likely to target JMJD3 in cancer as well as other pathologies

Lane 444: To our knowledge, GSK-J4 is the most studied of epi-drug concerning JMJD3.

Lane 475: a schema is added to conclude about the balance role of JMJD3 in carcinogenesis.

Lane 483: Abbreviations were added

CRISPRi

DDK1

EPHB4

Endo-siRNAs

lnRNAs

MCL1

ncRNAs

NOTCH2NL

piRNAs

PZLF

RPS6KA2

RUNX2

SNAI1

STAT3

TRA2A

U2AF1

XIAP

Please, to follow the right lane, refer to the word version with active track changes.

Moreover, our manuscript is ready to be correct to your MDPI English editing service.

Reviewer 1

  1. Lanes 33-35: Epigenetics is also influenced by intrinsic factors, which must be reflected in the statement.
  2. Lane 38: Epigenetic mechanisms are not restricted to non-coding microRNAs, what about long non-coding RNAs etc.?
  3. Each abbreviation should be explained in text, when used for the first time e.g. PTMs, MLL, HL etc.
  4. Lane 44, 176: the role of JMJD3 is not suppressing but demethylation
  5. Lane 59: actors or factors?
  6. Lane 65-67: the sentence is not correct
  7. Lane 78: the reference does not support the statement exactly
  8. Lane 79: too many “repeat”
  9. Lane 82: what do you mean by “trademark registered”
  10. Lane 111-113: The sentence should be somewhat introduced
  11. Lane 116: proteins?
  12. Lane 135: Correct the sentence
  13. Lane 171: Reference 48 shows slightly different age and %
  14. Lane 178-181: the sentence is confusing
  15. Lane 201: Reference 8 refers reference 53
  16. Lane 208: Later on what?
  17. Lane 225: Reference 62 = 55!
  18. Lane 244-245: rather repressive marks?
  19. Lane 254-257: the sentence is confusing
  20. Lane 263-265: It is not clear to me: JMJD3 deregulates AP-1 (dimer of fos and jun) and in parallel increases the expression of Fos and Jun?
  21. Lane 277: prognosis?
  22. Lane 278-280: the sentence is not completed
  23. Lane 282: There should be reference 72
  24. Lane 291-293: Check the sentence please
  25. Lane 322-324: The sentence is confusing
  26. Lane 351-354: Add reference
  27. Lane 367-370: what do you mean by the expression of JMJD3 in serum?
  28. Table 1: Ovarian: use “Cell death”
  29. Lane 379-380: the sentence is not completed
  30. Lane 381: I do not think that GSK-J4 is the most studied molecule of all.
  31. Lane 395-396: which team?

Lane 33-36: Epigenetics are influenced by external or internal environmental factors, for example, the specific characteristics of the cell, and plays a major role in oncogenesis by modulating chromatin compaction and consequently the expression of some genes involved in tumor development [2].

Lane 39-43: other small RNAs action were added.

action of non-coding RNAs (ncRNAs) such as small RNAs (micro RNAs (miRNAs), PIWI-interacting RNAs (piRNAs) and endogenous short RNAs (endo-siRNAS) or long non-coding RNAs (lnRNAs)) [3], DNA methylation and covalent post-translational modification (PTMs) of histones, among which acetylation and methylation are the most studied. Epigenetic modifications contribute to changes in chromatin compaction and are widely studied in cancerology [4–6].

HL abbreviation is at the right position lane 326. PTMs abbreviation has been modified lane 42.

Lane 48: suppress replaced by demethylate

Lane 63: actors replaced by factors

Lane 69-71: The sentence was clarified.

 Hypoacetylation and hypermethylation of histone tails are often associated with chromatin closure and low transcriptional activity, while histone tail hyperacetylation and hypomethylation are usually associated with chromatin opening and high transcription of genes

Lane 74-76: re writed sentence to clarify

Lane 84: a reference was changed

Lane 83-88: re writed sentence to clarify and not repeat

Lane 89: trademark registered modified by mark deposited

Lane 100: DDK1 abbreviation was added

Lane 107: PLZF abbreviation was added

Lane 111: RUNX2 abbreviation was added

Lane 121 : the sentence was introduced

Lane 125: LPS are not protein but lipopolysaccharides acting during inflammation. This was modify.

Lane 144-146: re writed sentence as JMJD3, a balance between oncogenic or tumor suppressor activity?

Lane 148: Glioblastoma for title

Lane 161: added Clustered Regularly Interspaced Short Palindromic Repeats Interference before (CRISPRi)

Lane 164: SNAI1 abbreviation was added

Lane 173: STAT3 abbreviation was added

Lane 183-187: sentences were clarify in removing the referring to different authors (Sherry Lynes and Ene et al.) and a conclusion was added.

Lane 190: medulloblastoma for title

Lane 191-193: changed 3 to 10 in 0 to 14 according the reference

Lane 198: changed suppresses to demethylate

Lane 201-205: Sentence was clarify.

. Moreover, the Shh MB subtype has a low H3K27me3 accumulation suggesting that JMJD3 is more active in tumors compared to normal cerebellar tissues and JMJD3 demethylase activity  inhibition by shRNA in Shh MB subtypes leads to a decrease in CGNPs proliferation and thus a decrease in growth MB [51].

Lane 214: add Neuroblastoma for title

Lane 230: suppressed reference 9

Lane 238: later on replaced by moreover

Lane 255: references was corrected. There was a problem in Zotero.

Lane 256 -259: MGMT U2AF1 TRA2A and RPS6KA2 abbreviations were added

Lane 278 : markers replaced by marks

Lane 282: Abbreviation HL was here

Lane 288-289: It was precise that NOTCH1 is implicated in oncogenic pathways

Lane 288-295: the sentence has been clarify: Ubiquitin-specific peptidase 7 (USP7) stabilizes the complex NOTCH1/JMJD3 through deubiquitination. The NOTCH1 oncogenic targets are controlled by this mechanism in controlling demethylation of target genes through JMJD3 activity. It was demonstrated that concomitant USP7 and JMJD3 inhibition conducted to a reduction of tumor growth in vivo.

Lane 301-308: information was added to better understand with a conclusion:

As in T-ALL cells, JMJD3 is overexpressed in HSCs and JMJD3 loss leads to upregulate the transcription factor AP-1 and increases the downstream expression of Jun and Fos resulting in an inability of HSCs to self-renew after stress. In a T-ALL model transduced with a retrovirus expressing NOTCH1 intra-cellular domain, authors showed that JMJD3 loss did not produce greater survival unlike AML model. The initiation and maintenance of T-ALL and AML by JMJD3 would depend on cell subtype and developmental context. JMJD3 could be an interesting target if its inhibition does not affect normal HSCs but only those implicated in leukemia [70].

Lane 314-316: Conclusion added: thus favoring leukemogenesis and justify of the predominant JMJD3 role in the repression of AML progression

321-323: The sentence is completed with : 

 Illiano et al, demonstrate a link between cAMP response element-binding protein (CREB), which plays a critical role in leukemogenesis, and JMJD3 by using GSK-J4 treatment on AML cell lines

Lane 325-327: Information about anti proliferative effect on AML cells were added: Moreover, it seems that the association of GSK-J4 with a natural cAMP raising compound (forskolin) potentiates the anti-proliferative effects of GSK-4 via PKA in some AML cells (U937 cells) leading to a BCL2 downregulation and caspase 3 activation

Lane 319: Diagnosis replaced by prognosis

Lane 321-323: the sentence was completed: Illiano et al, demonstrate a link between cAMP response element-binding protein (CREB), which plays a critical role in leukomogenesisleukemogenesis, and JMJD3 by using GSK-J4 treatment on AML cell lines.

Lane 325 : the reference was changed

Lane 326 : lpm1 corrected in lmp1

Lane 333: NOTCH homolog 2 N-terminal-like were added before NOTCH2NL

Lane 353: anaplastic large cell lymphomas (ALCLs) were added

Lane 356-358: conclusion added: JMJD3 is found to be a partial contributor to the activation of p16INK4a/Rb in an NPM-ALK context and could be a novel agent to contribute to upregulate the p16INK4a/Rb pathway

Lane 369-373: The sentence was clarify: Not surprisingly, as in glioma [43], JMJD3 also regulates tumor immunity in CRC by turning the transcription of Th-1 type chemokines (CXCL9, CXCL10) permissive, whereas in CRCand treated primary cells of CRC with GSK-J4 leads to a reduce in chemokines level. These chemokines have been shown to be positively correlated with CRC patients’ survival , chemokines are diminished [84].

Lane 377-380: Ephrin type-B receptor 4 added before  (EPHB4 )and EBPH4 corrected by EPHB4 (lane 380 also)

Lane 384 to 388: sentence has been clarify: Indeed, estrogen-resistant cancer cells overexpress HER2 that result in a constitutive activation of BCL2 by inactivating EZH2 through AKT pathway. In this case, the involvement of JMJD3 is no longer needed and this highlight the existence of non-genomic pathway in the regulation of BCL2 gene whereas JMJD3 is able to positively regulate BCL2 expression in a genomic way in non-resistant breast cancer cell lines  

Lane 392-396: Information have been added to clarify and conclude

Lane 398: It is precise that Oct4 is a stemness marker

Lane 410: reference 90 was add

Lane 398-400: A conclusion has been add about stemness characteristics in breast cancer.

In both luminal and triple-negative models, JMJD3 act in disfavor of stemness characteristics.

412-419: Sentences were added to clarify and conclude

 While Wei et al. demonstrated that a loss of H3K27 trimethylation is associated with a poor predictor in BC [93], another team highlighted the link between hypoxia, JMJD3 and the development of stem cell capacity by regulating the methylation state of DICER. The lack of oxygen leads to a reduction of JMJD3 oxygen-dependent demethylase activity that reduces DICER expression because of its over-methylated state. DICER is controlling miRNA production and DICER repression consequently conducts to a repression of miR-200 family that activates EMT [92,94]. In this context, JMJD3 activity seems to be important to counter the development of stem cell capacity and EMT thus reduces metastasis.

Lane 427: it is precise that the serum samples come from blood samples and lane 429 it was precised that authors looked JMJD3 mRNA expression in serum samples.

Lane 428 to 430: In response to the question “what do you mean by the expression of JMJD3 serum: I precise JMJD3 mRNA lane 429. Authors worked with blood samples and centrifuged samples to obtain serum. Analysis have been done on JMJD3 mRNA expression in serum samples.

Lane 437: Table 1 has been modified with all type of cancer event those described in the text.

Table 1: Cell Death replaced death cell

Lane 342-343: the sentence was completed: We list here in the table below (Table 2.) a few molecules likely to target JMJD3 in cancer as well as other pathologies

Lane 444: To our knowledge, GSK-J4 is the most studied of epi-drug concerning JMJD3.

Lane 475: a schema is added to conclude about the balance role of JMJD3 in carcinogenesis.

Lane 483: Abbreviations were added

CRISPRi

DDK1

EPHB4

Endo-siRNAs

lnRNAs

MCL1

ncRNAs

NOTCH2NL

piRNAs

PZLF

RPS6KA2

RUNX2

SNAI1

STAT3

TRA2A

U2AF1

XIAP

Reviewer 2 Report

The authors of the review ijms-1025946 entitled „The functions of the demethylase JMJD3 in cancer” provide a summary of the function and involvement of the demethylase JMJD3 in various tumor entities. They particularly focus on the oncogenic and tumor suppressive functions of JMJD3.

Altogether, a comprehensive review focusing on its different functions in different tumor entities might be useful for scientists. However, there are some major points that have to be addressed before publication.

Major comments:

First of all, the review is very hard to read due to unsuitable and confusing language style. In addition, there are some spelling mistakes, e.g. Table 2: Mecanisms, Table 1: Up-regulated, not Up regulated (the hyphen is not mandatory) etc.. The manuscript should be linguistically improved.

Often a final conclusion after giving many facts is missing. For example in line 340 – 342 the authors write: “The link between the modification of BCL2's chromatin architecture and its constitutive activation in estrogen-resistant cancer cells has been demonstrated, identifying JMJD3 and methyltransferase EZH2 as interesting therapeutic targets [87].” Firstly, EZH2 has not yet been mentioned in relation to breast cancer. Second, EZH2 fulfills the opposite function of JMJD3 and it remains unclear how EZH2 could serve as therapeutic target in this context. This is just an example, but throughout the review the authors give many examples and facts which the reader will only understand with detailed knowledge in the field because a conclusion or summary is missing. A review, however, should be easy to read, especially for less experienced scientist. To give one more example: line 262-266 - many facts but lack of a conclusion. The authors should carefully revise the text to link the given facts and information.

Existing clinical trials that target JMJD3 should be addressed, especially since the authors refer to epi-drugs.

The quality might be increase by adding a summarizing table for the function of JMJD3 in all and not only in cancer types which are not explained in the text. Also a figure that depicts the different mechanisms would be helpful. Furthermore, a table or figure showing and summarizing interactions with other factors/inhibitors would contribute to better understanding.

Minor comments:

There is growing evidence that JMJD3 functions independent on its demethylase activity. The authors should include this essential fact, especially when they talk about demethylase independent mechanisms, such as the regulation of ARF and Oct4.

It leads to confusion when the authors refer to a reference (...) et al. (...) (e.g. line 167 or 169 etc.) when it is unclear what exactly was meant without taking a closer look at the reference list.

Abbreviations not at the right position or not advertised, e.g. line 39.

Author Response

Please, to follow the right lane, refer to the word version with active track changes.

Moreover, our manuscript is ready to be correct to your MDPI English editing service.

Reviewer 2

“First of all, the review is very hard to read due to unsuitable and confusing language style. In addition, there are some spelling mistakes, e.g. Table 2: Mecanisms, Table 1: Up-regulated, not Up regulated (the hyphen is not mandatory) etc.. The manuscript should be linguistically improved.”

We tried to correct spelling mistakes, we are aware that English needs to be revised. We forward the manuscript to the MPDI English editing service.

“Often a final conclusion after giving many facts is missing. For example in line 340 – 342 the authors write: “The link between the modification of BCL2's chromatin architecture and its constitutive activation in estrogen-resistant cancer cells has been demonstrated, identifying JMJD3 and methyltransferase EZH2 as interesting therapeutic targets [87].” Firstly, EZH2 has not yet been mentioned in relation to breast cancer. Second, EZH2 fulfills the opposite function of JMJD3 and it remains unclear how EZH2 could serve as therapeutic target in this context. This is just an example, but throughout the review the authors give many examples and facts which the reader will only understand with detailed knowledge in the field because a conclusion or summary is missing. A review, however, should be easy to read, especially for less experienced scientist. To give one more example: line 262-266 - many facts but lack of a conclusion. The authors should carefully revise the text to link the given facts and information.”

We tried to clarify some sentences and had conclusions:

Lane 390-398 : The link between the modification of BCL2's chromatin architecture and its constitutive activation in estrogen-resistant cancer cells has been demonstrated, identifying JMJD3 and methyltransferase EZH2 as interesting therapeutic targets. Indeed, estrogen-resistant cancer cells overexpress HER2 that results in a constitutive activation of BCL2 by inactivating EZH2 through AKT pathway. In this case, the involvement of JMJD3 is no longer needed and this highlight the existence of non-genomic pathway in the regulation of BCL2 gene by EZH2 whereas JMJD3 is able to positively regulate BCL2 expression in a genomic way in non-resistant breast cancer cell lines [88]. JMJD3 could be targeted in nonresistant cancer cells and EZH2 in both nonresistant cancer cells and estrogen-resistant cancer cells, to limit BCL2 activation.

Lane 288-308 : . NOTCH1 is implicated in oncogenic pathways. In 2014, JMJD3 has been identified as a central player in oncogene activation mediated by NOTCH1. JMJD3 was found to be overexpressed and colocalized with NOTCH1 in T-ALL cells. Ubiquitin-specific peptidase 7 (USP7) stabilizes the complex NOTCH1/JMJD3 through deubiquitination. The NOTCH1 oncogenic targets are controlled by this mechanism in controlling demethylation of target genes through JMJD3 activity. It was demonstrated that concomitant USP7 and JMJD3 inhibition conducted to a reduction of tumor growth in vivo. [67]. JMJD3 was found at the promotor site of NOTCH1 targets that have oncogenic functions HES1, HEY1, NRARP [68], c-myc and NOTCH3 [69]. It allows T-ALL cells initiation and maintenance by controlling methylation of oncogenic targets [68]. These studies suggest that JMJD3 is an interesting therapeutic target to reduce T-ALL progression. Other researchers have examined JMJD3 functions in hematopoietic stem cells (HSCs) that is not related to chromatin modification. As in T-ALL cells, JMJD3 is overexpressed in HSCs and JMJD3 loss leads to upregulate the transcription factor AP-1 and increases the downstream expression of Jun and Fos resulting in an inability of HSCs to self-renew after stress. In a T-ALL model transduced with a retrovirus expressing NOTCH1 intra-cellular domain, authors showed that JMJD3 loss did not produce greater survival unlike AML model. The initiation and maintenance of T-ALL and AML by JMJD3 would depend on cell subtype and developmental context. JMJD3 could be an interesting target if its inhibition does not affect normal HSCs but only those implicated in leukemia [70].

“It leads to confusion when the authors refer to a reference (...) et al. (...) (e.g. line 167 or 169 etc.) when it is unclear what exactly was meant without taking a closer look at the reference list.”

Lane 183-187: sentences were clarify in removing the referring to different authors (Sherry Lynes and Ene et al.) and a conclusion was added.

Lane 144-146: re writed sentence as JMJD3, a balance between oncogenic or tumor suppressor activity?

Lane 201-205: Sentence was clarify.

. Moreover, the Shh MB subtype has a low H3K27me3 accumulation suggesting that JMJD3 is more active in tumors compared to normal cerebellar tissues and JMJD3 demethylase activity  inhibition by shRNA in Shh MB subtypes leads to a decrease in CGNPs proliferation and thus a decrease in growth MB [51].

Aptamer treatment also resulted in an increase in P53 in response to increased JMJD3 expression [49]. Thus, result contributes to place JMJD3 in a central position between STAT3 and p53 to limit tumor progression.

Lane 314-316: Conclusion added: thus favoring leukemogenesis and justify of the predominant JMJD3 role in the repression of AML progression

321-323: The sentence is completed with : 

 Illiano et al, demonstrate a link between cAMP response element-binding protein (CREB), which plays a critical role in leukemogenesis, and JMJD3 by using GSK-J4 treatment on AML cell lines

Lane 325-327: Information about anti proliferative effect on AML cells were added: Moreover, it seems that the association of GSK-J4 with a natural cAMP raising compound (forskolin) potentiates the anti-proliferative effects of GSK-4 via PKA in some AML cells (U937 cells) leading to a BCL2 downregulation and caspase 3 activation

Lane 321-323: the sentence was completed: Illiano et al, demonstrate a link between cAMP response element-binding protein (CREB), which plays a critical role in leukomogenesisleukemogenesis, and JMJD3 by using GSK-J4 treatment on AML cell lines.

Lane 356-358: conclusion added: JMJD3 is found to be a partial contributor to the activation of p16INK4a/Rb in an NPM-ALK context and could be a novel agent to contribute to upregulate the p16INK4a/Rb pathway

Lane 369-373: The sentence was clarify: [81]. Not surprisingly, as in glioma [43], JMJD3 also regulates tumor immunity in CRC by turning the transcription of Th-1 type chemokines (CXCL9, CXCL10) permissive, whereas in CRCand treated primary cells of CRC with GSK-J4 leads to a reduce in chemokines level. These chemokines have been shown to be positively correlated with CRC patients’ survival , chemokines are diminished [84].

Lane 398-400: A conclusion has been add about stemness characteristics in breast cancer.

In both luminal and triple-negative models, JMJD3 act in disfavor of stemness characteristics.

412-419: Sentences were added to clarify and conclude

 While Wei et al. demonstrated that a loss of H3K27 trimethylation is associated with a poor predictor in BC [93], another team highlighted the link between hypoxia, JMJD3 and the development of stem cell capacity by regulating the methylation state of DICER. The lack of oxygen leads to a reduction of JMJD3 oxygen-dependent demethylase activity that reduces DICER expression because of its over-methylated state. DICER is controlling miRNA production and DICER repression consequently conducts to a repression of miR-200 family that activates EMT [92,94]. In this context, JMJD3 activity seems to be important to counter the development of stem cell capacity and EMT thus reduces metastasis.

Lane 74-76: re writed sentence to clarify

The constant "cross-talk" between the different epigenetic marks positioned in the genome at a given time leads to a real complexity in the understanding of epigenetic modifications. This in turn has a different impact on gene expression, integrity and stability of the genome

The quality might be increase by adding a summarizing table for the function of JMJD3 in all and not only in cancer types which are not explained in the text. Also a figure that depicts the different mechanisms would be helpful. Furthermore, a table or figure showing and summarizing interactions with other factors/inhibitors would contribute to better understanding.

Lane 437: Table 1 has been modified with all type of cancer event those described in the text.

Lane 475: a schema is added to conclude about the balance role of JMJD3 in carcinogenesis.

“Existing clinical trials that target JMJD3 should be addressed, especially since the authors refer to epi-drugs.” :

To our knowledge, there is no clinical trial that target JMJD3. Studies presented in this review are speaking about fundamental research.

Lane 474: it was precised that epidrug could be important for clinical trial to treat patients in the future.

“There is growing evidence that JMJD3 functions independent on its demethylase activity. The authors should include this essential fact, especially when they talk about demethylase independent mechanisms, such as the regulation of ARF and Oct4.”

It is a choice to not write another paragraph about JMJD3 functions independent on its demethylase activity to not repeat and because it is clearly precise in the main text when JMJD3 acts independently on its demethylase activity.

“Abbreviations not at the right position or not advertised, e.g. line 39.”

PTMs abbreviation has been modified lane 42.

Lane 100: DDK1 abbreviation was added

Lane 107: PLZF abbreviation was added

Lane 111: RUNX2 abbreviation was added

Lane 161: added Clustered Regularly Interspaced Short Palindromic Repeats Interference before (CRISPRi)

Lane 164: SNAI1 abbreviation was added

Lane 173: STAT3 abbreviation was added

Lane 256 -259: MGMT U2AF1 TRA2A and RPS6KA2 abbreviations were added

Lane 377-380: Ephrin type-B receptor 4 added before  (EPHB4 )and EBPH4 corrected by EPHB4 (lane 380 also)

Lane 483: Abbreviations were added

CRISPRi

DDK1

EPHB4

Endo-siRNAs

lnRNAs

MCL1

ncRNAs

NOTCH2NL

piRNAs

PZLF

RPS6KA2

RUNX2

SNAI1

STAT3

TRA2A

U2AF1

XIAP

Round 2

Reviewer 1 Report

The manuscript has been substantially improved, however, minor inaccuracies are still there e.g. double numbering in References, Lane 331: the sentence is incorrect etc. The overall manuscript should be checked by native speaker prior its acceptance for publication.   

Author Response

Reviewer 1

By referring to the file with the "single mark" modification tracking

“The manuscript has been substantially improved, however, minor inaccuracies are still there e.g. double numbering in References, Lane 331: the sentence is incorrect etc. The overall manuscript should be checked by native speaker prior its acceptance for publication.”

Lane 338: “double actor” replace “double-hatted actor”

Final Figure (Schema 1.) has been suppressed according to Reviewer 2’s advice.

The manuscript was transmitted and corrected by MDPI English editing service.

Reviewer 2 Report

The authors of the review ijms-1025946 entitled „The functions of the demethylase JMJD3 in cancer” successfully revised their manuscript. Especially the extension of table 1 provides a good overview and contributes to the quality of the review. After final correction by the MDPI English editing service, I recommend the manuscript for publication in Int. J. Mol. Sci.. However, I would like to make one final comment: The new figure is not necessary in view of the extension of Table 1 if the exact mechanism or mediating key factors or circumstances are not included in the figure.

Author Response

By referring to the file with the "single mark" modification tracking

“The authors of the review ijms-1025946 entitled „The functions of the demethylase JMJD3 in cancer” successfully revised their manuscript. Especially the extension of table 1 provides a good overview and contributes to the quality of the review. After final correction by the MDPI English editing service, I recommend the manuscript for publication in Int. J. Mol. Sci.. However, I would like to make one final comment: The new figure is not necessary in view of the extension of Table 1 if the exact mechanism or mediating key factors or circumstances are not included in the figure.”

Final Figure (Schema 1.) has been suppressed.

The manuscript was transmitted and corrected by MDPI English editing service.
